# Haitian coffee agroforestry systems harbor complex arabica variety mixtures and under-recognized genetic diversity

Claude Patrick Millet[1,2,3,4]*, Clémentine Allinne[3,4,5,6], Tram Vi[1,7], Pierre Marraccini[1,8], Lauren Verleysen[9,10], Marie Couderc[1], Tom Ruttink[10,11], Dapeng Zhang[12], William Solano Sanchéz[13], Christine Tranchant-Dubreuil[1], Wesly Jeune[2,14], Valérie Poncet[1]*

1 IRD, UMR DIADE, CIRAD, Université Montpellier, Montpellier, France, 2 Faculté des Sciences de l'Agriculture et de l'Environnement, Université de Quisqueya, Port-au-Prince, Haiti, 3 Institut Agro, ABSys, Université Montpellier, CIHEAM-IAMM, CIRAD, INRAE, Montpellier, France, 4 CIRAD, UMR ABSys, F-34398, Montpellier, France, 5 GECO, Université Montpellier, CIRAD, Montpellier, France, 6 CIRAD, UPR GECO, F-34398, Montpellier, France, 7 Agricultural Genetics Institute (AGI), Hanoi, Vietnam, 8 CIRAD, UMR DIADE, Montpellier, France, 9 Faculty of Sciences, Division of Ecology, Evolution and Biodiversity Conservation, KU Leuven, Leuven, Belgium, 10 ILVO, Melle, Belgium, 11 Ghent University, Ghent, Belgium, 12 USDA-ARS, SPCL, Beltsville, Maryland, United States of America, 13 CATIE, Turrialba, Costa Rica, 14 AVSF, Pétion-Ville, Haïti

* claudepatrickmillet@gmail.com (CPM); valerie.poncet@ird.fr (VP)

**Data Availability Statement:** All relevant data are available within the paper and its Online Resource files. The passport and genotyping (KASP SNP genotypes and HiPlex amplicon sequencing

## Abstract

Though facing significant challenges, coffee (*Coffea arabica*) grown in Haitian agroforestry systems are important contributors to rural livelihoods and provide several ecosystem services. However, little is known about their genetic diversity and the variety mixtures used. In light of this, there is a need to characterize Haitian coffee diversity to help inform revitalization of this sector. We sampled 28 diverse farms in historically important coffee growing regions of northern and southern Haiti. We performed KASP-genotyping of SNP markers and HiPlex multiplex amplicon sequencing for haplotype calling on our samples, as well as several Ethiopian and commercial accessions from international collections. This allowed us to assign Haitian samples to varietal groups. Our analyses revealed considerable genetic diversity in Haitian farms, higher in fact than many farmers realized. Notably, genetic structure analyses revealed the presence of clusters related to Typica, Bourbon, and Catimor groups, another group that was not represented in our reference accession panel, and several admixed individuals. Across the study areas, we found both mixed-variety farms and monovarietal farms with the historical and traditional Typica variety. This study is, to our knowledge, the first to genetically characterize Haitian *C. arabica* variety mixtures, and report the limited cultivation of *C. canephora* (Robusta coffee) in the study area. Our results show that some coffee farms are repositories of historical, widely-abandoned varieties while others are generators of new diversity through genetic mixing.

haplotypes) data that support the findings of this study are available in the DataSuds repository (IRD, France) at https://doi.org/10.23708/T6YZML.

**Funding:** C.P.M. was funded by PhD grants from the French Embassy in Haiti and the ARTS program (IRD). This work was supported by the Agricultural and Agroforestry Technological Innovation Program (HA-L1107, HA-G1038) funded by the IDB, GAFSP, IFAD and Haitian government. T.V. was supported by PhD grants from the French Embassy in Vietnam and the ARTS program (IRD). L.V. was supported by Research Foundation-Flanders (FWO; G090719N). The funders had no role in study design, data collection and analysis, decision to publish, or preparation of the manuscript.

**Competing interests:** The authors have declared that no competing interests exist.

# 1 Introduction

## 1.1 Diversity and resilience of agrosystems

Agricultural systems are faced with various stressors, both biotic (pests, diseases) and abiotic (droughts, nutrient deficiencies, temperature shocks, etc.). Modern agricultural practices for responding to these pressures are increasingly called into question for their unsustainable nature, especially as these issues are exacerbated by global climate change [1]. There is thus ever-growing urgency to improve the resilience and sustainability of crop systems and the various services they provide. Crop diversity in farming systems can help this endeavor by supporting the delivery of ecosystem services without compromising on productivity [2,3]. Diversification of cropping systems can be achieved at the farm scale by combining multiple species [4], but varietal diversity at the crop level can also contribute to crop productivity and resilience [5]. For instance, there is evidence that intraspecific diversity can suppress pests and pathogens by increasing spatial and genetic heterogeneity and decreasing the proportion of susceptible individuals [6,7]. It can also increase yield stability, particularly under stress [8]. Consequently, there is increased scientific interest in the effect of varietal mixtures, particularly for annual crops such as cereals [9,10].

## 1.2 Agroforestry systems

The benefits of agrobiodiversity are particularly well-mobilized in diversified agroforestry systems, especially smallholder farms and homegardens in tropical regions. These often-traditional systems feature considerable botanical richness, combining several cash and subsistence crops with fuelwood, timber, medicinal and culturally important species [11,12]. As such, they are important providers of several ecosystem services such as food security, farmer economic resilience, pest and disease regulation, nutrient cycling, soil conservation, preservation of local knowledge and traditions, as well as conservation of natural biodiversity and crop genetic diversity [13,14].

This role is of even greater importance in countries such as Haiti, which is faced with the heavy degradation of its natural ecosystems and their biodiversity, and therefore the risk of losing associated ecosystem services [15,16]. In fact, most of the country's remaining forested areas are primarily agroforestry systems [17]. Furthermore, Haiti is a Least Developed Country whose population is primarily rural. The economic resilience provided by diversified agroforestry systems is therefore valued by farmers [18,19]. These so-called "Creole Gardens" combine fruit, timber and charcoal trees with perennial crops such as coffee, cacao, banana and plantain, as well as annual crops such as yam and taro [20]. Ensuring the continued social, economic and ecological viability of these systems is thus of prime importance. The present study focuses on Haitian Coffee Agroforestry Systems (hereafter "CAFS").

## 1.3 *Coffea arabica*: Biology and domestication

*Coffea arabica* L. **(Rubiaceae)** is an allotetraploid (2n = 4x = 44), amphidiploid, mainly autogamous, fairly recent (665,000 years ago) natural hybrid of diploid ancestors species *C. canephora* and *C. eugenioides* [21,22]. It is the most important species for the global coffee beverage market [23] representing about 60% of the global production [24].

Though originating from the mountains of Ethiopia, *C. arabica* was first cultivated in Yemen, and became widespread in tropical regions in the XVII-XVIII[th] centuries as it was introduced to various European colonies. Two main lineages of *C. arabica* were cultivated outside of Yemen: the Typica and Bourbon lines, which both experienced strong successive

genetic bottlenecks leading to a much narrower genetic diversity compared to natural populations [25,26].

Despite this low diversity, several cultivated varieties were developed in the XX[th] century onwards, taking advantage of mutations in the two main lines. These "modern" varieties were later crossed with Ethiopian accessions or the Timor Hybrid (a spontaneous hybrid of *C. arabica* and *C. canephora* [27,28] to introduce traits of agronomic interest such as pest and disease resistance [29]. Currently, *C. arabica* is grown in a variety of agricultural systems ranging from full-sun intensive monoculture to shaded, botanically and structurally complex agroforestry systems [30]. As for many crops, global climate change is expected to severely and negatively affect Arabica coffee cultivation [31,32] but increasing the proportion of shade-grown coffee (and therefore CAFS) can mitigate that impact [33].

## 1.4 Opportunities provided by coffee genetic diversity

As adaptation to environmental changes and stresses can be achieved through mobilization of crop genetic diversity, germplasm collections are an important contributor to the continued vitality of the coffee sector. The CATIE international collection is one of the better-known Arabica collections [34,35], with a total of 1975 accessions, including Ethiopian wild materials, traditional and modern varieties, as well as more recent hybrid lines bred for quality, productivity, and disease resistance. However, *ex-situ* conservation of genetic resources can have certain drawbacks: it effectively freezes the process of natural and farmer selection, preventing adaptation to changing conditions in the field [36]. At the same time, collected accessions are not truly static either, but may lose their genetic integrity over time through outcrossing or identification errors [37,38]. The more dynamic *in-situ* (farm-based) conservation of genetic diversity can therefore serve as a beneficial complementary approach. Leveraging diversity for adaptation and improvement of coffee yield, quality and resilience has been made easier through the increasing availability of marker-based methods of molecular characterization of coffee plant genotypes [21,22,25,39]. Due to *C. arabica* being a relatively recent species with low genetic diversity made even lower in cultivated accessions by successive bottlenecks, its genome displays a low level of polymorphism. Therefore, targeted genotyping of regions with known polymorphisms, and particularly Single Nucleotide Polymorphisms (SNPs) has shown promise as a cost-effective and efficient way to characterize and identify the genetic identity, diversity and population structure of Arabica coffee [22,40–42]

## 1.5 Coffee in Haiti

Coffee cultivation on Haitian soil dates back to the French colonial era, when the Typica line was introduced and established on the mountainous island in 1735 [43]. By the end of the XVIII[th] century, the then-colony of St-Domingue had become the first coffee producer in the world, accounting for half of the global supply [23,44]. When Haiti became independent in 1804, *C. arabica* 'Typica' remained an important agricultural resource, though multiple crises throughout the country's history reduced its productivity and contribution to the nation's economy [45,46]. Examples include a lack of financial and technical support to farmers, world market volatility, coffee pest (e.g. the coffee berry borer, *Hypothenemus hampei*) and disease (e.g. Coffee Leaf Rust, *Hemileia vastatrix*) outbreaks, soil erosion and nutrient loss, and coffee stand aging [47].

In the present day, these issues are still ongoing, but Haitian coffee retains the potential to be attractive on smaller-batch, specialty markets. Indeed, Creole Gardens are the main type of coffee-cropping systems and receive virtually no agrochemical inputs [48]. They can produce ethical and environmentally-friendly shade coffee, which is of high quality [49]. Some growers'

cooperatives are able to sell their harvest on North-American, European and Japanese gourmet markets at high prices [47,48], and local workshops with growers suggest a renewed interest in reviving this once-important culture [50]. The Haitian context makes clear that any effort to revitalize local coffee production must do so in a way that safeguards the ecological value of diversified CAFS and their contribution to the socioeconomic resilience and food security of local communities.

## 1.6 Study aims

With few publications about Haitian coffee in the scientific literature, reports from multilateral development and NGO projects remain the main source of information. These suggest that Typica remains the main variety grown in Haitian fields, but that from the 1970s onwards new varieties were introduced by several agricultural development programs [48,51,52]. The distribution and dissemination of coffee varieties in Haitian CAFS is poorly documented. The revitalization of diverse agroforestry systems is of economic, social, cultural and ecological importance. Therefore, there is a need to better understand the genetic diversity contained therein, as well as its structure, in order to assist in the decision-making of farmers and policy actors and to help conserve coffee genetic resources.

This study therefore aimed to characterize the genetic diversity and genetic structure of Arabica coffee cultivated in Haitian CAFS. We focused on collecting data from two administrative departments of development priority: Nord and Grande-Anse, in northern and southern Haiti, respectively. In each department, 14 farms were selected to collect leaf samples for genetic analysis and information on the coffee tree was recorded through farmer surveys. The objectives were: (i) to determine varietal diversity and its distribution over the territory using SNP target markers and a set of Arabica diversity reference accessions, (ii) to compare local knowledge of this diversity based on farmer survey to genetic variety classification; (iii) to draw up guidelines for the dynamic conservation and development of this varietal diversity.

## 2 Materials and methods

### 2.1 Selection of study sites

The present study took place in two administrative departments, Nord and Grande-Anse, in northern and southern Haiti, respectively. These regions were historically important coffee producers since the colonial era [44], but now struggle to sustain production. Sampling sites were selected on the basis of a preliminary survey of 122 Non-intensive, diversified CAFS, of which 43 were subsequently surveyed in-depth. These surveys were conducted in 2021 by the multilaterally-funded Agricultural and Agroforestry Technological Innovation Program (PITAG) implemented by *Agronomess et Vétérinaires Sans Frontières* and the Haitian Ministry of Agriculture. Twenty-eight of the surveyed CAFS (14 per department) were selected on the basis of farmer-reported varietal diversity for inclusion in the present study. Selection of study sites took into account geographical spread, range of farmer-reported (expected) varietal composition, and inclusion of several municipalities in order to attain good representation of diversified Haitian CAFS in the two administrative departments. Selected gardens were thus spread across five municipalities (*communes*): Bahon, Dondon and Grande Rivière du Nord (Nord); Beaumont and Pestel (Grande-Anse) (S1 Table). After in-person explanation of the study aims, signatures on Prior Informed Consent (PIC) forms were gathered from all farmers. Material transfer agreements (MTAs) and export permits for the collected samples were obtained from the Haitian Ministry of Agriculture, Natural Resources and Rural Development (MARNDR).

## 2.2 Sampling strategy and local knowledge collection

Coffee leaf samples were collected in November-December 2021. In order to capture the diversity present in the surveyed gardens, farmers were asked to identify the various (putative) coffee varieties growing in their field. At each site but one, a minimum of 20 plants were sampled, taking care to include all putative varieties in similar quantities. In addition, farmers were asked to point out any and all coffee plants that seemed atypical to them, for example in their health, vigor, productivity, or lack thereof. When such coffee trees were identified, they were sampled as well. Therefore, our sampling covered a wide range of coffee tree ages, phenotypes and putative varieties. Four healthy, mature leaves from plagiotropic axes of the coffee trees were collected, when possible from the third fully-grown pairs from the apex, and preserved in silica gel.

In addition to the Haitian field samples, a panel of varieties and wild accessions (n = 96) was obtained from the CATIE international coffee germplasm collection (Turrialba, Costa-Rica) with additional samples (n = 15) from the HARC collection (Hawaii, USA) which were provided by USDA-ARS, SPCL (Beltsville, MD, USA) to serve as references for varietal assignment. Finally, wild *C. arabica* (n = 6), as well as *C. canephora* (n = 5), *C. liberica* (n = 3) and *C. congensis* (n = 2) were sampled from the IRD collection (Montpellier, France) as outgroups (S2 Table).

## 2.3 Vernacular names

Any work taking advantage of Haitian coffee diversity will need to understand how its stewards perceive and categorize it in order to maximize the effectiveness of farmers' participation. Putative varietal identification of collected field samples were recovered from surveys. These came primarily from farmers, although in some cases this was impossible and the identification came from local agronomists instead. Taken together, these identifications (hereafter "vernacular categories") were considered to provide an indication of local knowledge. To avoid confusion, vernacular category names were written with the Haitian creole spelling. When no clear identification could be provided, a "No ID" record was made for that sample.

## 2.4 Molecular method

**2.4.1 KASP SNP genotyping.** A set of 96 SNPs were first genotyped by KASP assay. Mérot-L'Anthoëne et al. [39] developed a DNA array of 8580 biallelic SNP from *C. canephora* and *C. arabica* sequencing, 945 of which were designed specifically to be informative for *arabica* diversity assessment. From the latter, Zhang et al. [53] selected a set of 96 core SNPs found to be polymorphic and discriminant across a panel of commercial and wild accessions of *C. arabica*, including many of our reference samples from the CATIE international collection. Selective genotyping by KASP assay targeting these core markers was conducted by LGC Biosearch Technologies (Middlesex, UK). Raw genotyping data (coded as A/B, biallelic markers) was filtered to exclude loci with missing data in <30% of samples, and individuals with <30% missing genotype data. Genotyping error rates were calculated using three duplicate samples included for genotyping, as percent difference in genotype calls between duplicates across all called SNP sites.

**2.4.2 HiPlex amplicon sequencing.** In parallel, genotyping data was obtained on the same sample set using highly multiplex amplicon sequencing (HiPlex) of 400 regions (110–120 bp each) selected to be variable between the Typica and Bourbon varieties (Bawin, 2022). DNA was extracted from Haitian and reference samples using a protocol with MATAB+DTT lysis buffer following a sorbitol wash of the material (adapted from [54]. Amplicon libraries were prepared by Floodlight Genomics LLC (Knoxville, TN, USA), and were sequenced by Admera

Health (South Plainfield, NJ, USA). Obtained reads were mapped onto each of the two subgenomes of the *C. arabica* reference genome sequence v0.6 of the accession ET-39 [26] using a customized script by Bawin (2020), available on GitLab (https://gitlab.com/ybawin/sequence-data-processing-tetraploids). After filtering, read-backed haplotyping was conducted based on SNPs in the HiPlex read data using the SMAP software package v4.2.0 [55]. The full pipeline description can be found in supplementary notes. Genotype tables were then filtered to exclude loci with missing data in >30% of Arabica samples specifically, then individuals of all *Coffea* species with >30% missing data. Variant calling error rates were calculated using 12 duplicate samples included for sequencing.

## 2.5 Genetic analyses and varietal characterization

Full genetic analyses were carried out on the targeted SNPs genotyped with the KASP assay, since they were optimized to reveal intra-Arabica (both between cultivated and wild accessions) diversity. Then, the results were compared with HiPlex haplotype data designed to distinguish Typica and Bourbon varieties and their potential hybrids.

Global genetic diversity and its geographical distribution were estimated by using descriptive statistics and genetic differentiation pairwise $F_{ST}$ values (with corresponding p-values at 999 bootstraps) calculated at the farm, municipality and department levels using GenAlEx software package v. 6.51b2 [56]. For the latter, values were calculated both per locus within groups and per group across all loci. The significance of each hierarchical level was tested using the varcomp.glob, as well as test.between, test.within and test.between.within (1000 permutations each) function in the R package Hierfstat v. 0.5–11 [57].

Varietal and farm composition characterization were carried out in several steps. Principal component analyses (PCA) was performed on Haitian and reference samples (both including and excluding reference samples) using the R packages LEA v. 3.10.2 [58] and Tidyverse v. 2.0.0 [59] ggplot function. Unweighted neighbor-joining dendrograms were made from a simple-matching distance matrix with 1000 bootstrap replicates using the software DARwin v. 6.0.21 [60] in order to visualize the diversity of Haitian coffee plants in relation to that of reference samples. In addition, an analysis of population structure using the sNMF function (K = 1–10, 100 repetitions) of the R package LEA v. 3.10.2 was conducted. A threshold of 80% membership to a sNMF population was used to assign samples to a varietal group (hereafter "genetic group"), which was labeled according to the reference samples in that group. Samples with <80% membership in any group were considered to be admixed. In order to infer putative intervarietal crosses and more fully describe the sampled coffee diversity, we used a threshold of 40% membership from one or more genetic groups to sub-categorize admixed individuals. Pairwise $F_{ST}$ (and p-values at 999 bootstraps) were calculated between genetic groups (including and excluding reference samples) with GenAlEx v. 6.51b2.

The results obtained with the targeted KASP SNP genotyping data were then compared with those obtained with HiPlex amplicon sequencing haplotype data designed to distinguish Typica and Bourbon varieties and their potential hybrids. In addition to the above analyses, a Mantel test of correlation was performed on the HiPlex haplotype and KASP SNP genotyping data distance matrices using GenAlEx v. 6.51b2. Allele counts for *C. arabica*-only (field and reference) HiPlex haplotype data were extracted using the adegenet R package v. 2.1.10 [61] and a sNMF analysis of population structure was conducted with the same parameters as for the KASP SNP genotyping data. Genetic groups were again defined using an 80% membership threshold. Farm genetic group composition was mapped in QGIS v. 3.30.1[62] using Natural Earth (Free vector and raster map data @ naturalearthdata.com) and shapefiles from Hijmans and UC Berkeley [63,64]and Patterson and Kelso [65].

For subsequent analyses and discussion of the results, the KASP-based genetic groups were retained as they were determined on the basis of markers selected using a broader panel of Arabica accessions [53], and more accessions were represented in the reference samples for the present study.

## 2.6 Vernacular and genetic identification comparison

We investigated the degree of correspondence between local *C. arabica* vernacular category names and their genotypic makeup from the KASP SNP genotyping data by testing for correlation between the sizes of the sNMF-determined genetic groups (with admixed individuals lumped together as one group, as well as sub-categorized according to the 40% membership threshold) and those of the vernacular categories (including the "No ID" group). We also tested for correlation between the number of varieties grown on farms (as reported by farmers) and the number of varietal clusters identified for that field, excluding admixed individuals. For all these analyses, we used a series of Pearson's $\chi^2$ tests of independence in R.

Farms were categorized based on the local perception of their diversity (with "diverse" farms having more than two reported vernacular categories and "less diverse" farms having one or two) and their genetic group diversity (again, with "diverse farms" having more than two genetic groups and "less diverse" farms having one or two). We tested the correlation between locally-perceived and genetic group diversity. The genetic diversity categories were the same regardless of whether the admixed individuals were sub-categorized. Another test was performed, this time categorizing farms as either "monovarietal"(only one group) or "multi-varietal" (two or more groups), again according to vernacular category and genetic group diversity. Pearson's $\chi^2$ tests of independence were again used for all these analyses.

## 3 Results

### 3.1 Genetic diversity analyses

After filtering, KASP SNP genotyping data were obtained for 724 individuals (including 117 *C. arabica*, five *C. canephora* and one *C. congensis* reference sample) at 87 loci. There were no differences over all loci for the three duplicated samples.

Over all Haitian samples, expected heterozygosity (or gene diversity, $H_e$) was 0.33, only slightly lower than among all Arabica references in the *ex-situ* collections ($H_e = 0.37$). Across farms, $H_e$ ranged from 0.02 (*Ka Gous*, farm code G13) to 0.34 (*Bertin*, farm code N07). Mean $H_e$ and marker polymorphism were lower across farms (0.23 and 66.7% respectively) than across municipalities (0.28 and 83.9%) and departments (0.32 and 89.7%). In all but the lowest diversity farms, observed heterozygosity ($H_o$) was consistently lower than $H_e$, as expected for autogamous species. The inbreeding coefficient ($F_{IS}$) of farms ranged from 0.38 to 0.85, with an intermediate value across Arabica reference accessions ($F_{IS} = 0.62$). These statistics, which are detailed in supplementary materials (S3–S5 Tables), indicate that there is considerable diversity across Haitian samples, but that farms exhibit great variability in the genetic makeup of their germplasm.

Statistical tests of the effect of different geographic levels on coffee genetic structure in Hierfstat revealed a non-significant impact of the department (p = 0.431) and municipality (p = 0.141) levels, which is also apparent in pairwise $F_{ST}$ comparisons (S6 Table). However, the municipality of Bahon (containing four farms) was notable in that it harbored less diversity ($H_o = 0.043$) than the single farm in the adjacent municipality of Grande Rivière du Nord ($H_o = 0.067$). The farm level was found to have a significant impact (p = 0.001) on genetic structure of sampled Haitian Arabica. Therefore, within- and between farm heterogeneity was the main source of variation in the genetic structure of coffee tree stands (S7 Table).

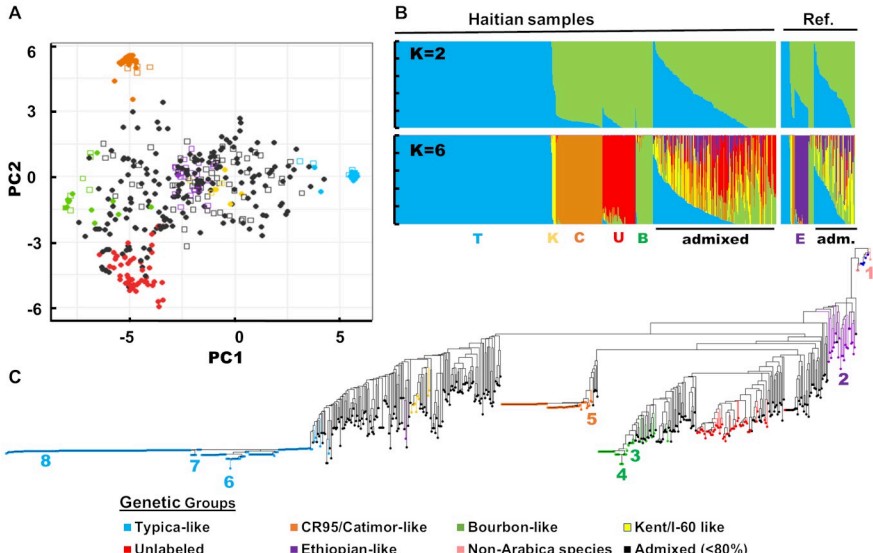

**Fig 1. Genetic structure and diversity of Haitian coffee in relation to reference samples from international collections, based on SNP genotyping data. A.** Principal Component Analysis performed on Haitian (filled circles) and reference (open squares) *C. arabica* samples showing the first two axes (46,89% and 11,26% of variance explained, respectively). **B.** *C. arabica* population structure analysis at K = 2 (top) and K = 6 (bottom) for Haitian and reference (Ref.) samples. Initials at bottom correspond to genetic groups defined in the legend, with only the Ethiopian-like and admixed (adm.) reference individuals labeled. **C.** Unweighted neighbor-joining dendrogram of reference and Haitian coffee samples (with non-*C. arabica* species in pink at the root) based on simple matching distance matrix. Numbers represent select reference individuals for illustrative purposes: 1) GUI2 *C. canephora*; 2) T.02731 Jimma Galla Sidamo; 3) T.02542 Caturra; 4) Ku042 Red Bourbon; 5) T.08867 CR-95; 6) T.03427 Cera; 7) Ku214 Jamaica Blue Mountain; 8) T.00989 Guadeloupe. Sample color code was based on 80% membership threshold in one ancestral population at K = 6 from population structure analysis.

The lack of genetic structure at department and municipality levels was also seen in the scattered distribution of Haitian samples along the main axes of the PCA (Fig 1A). Reference samples of *ex-situ* collections were also quite scattered, suggesting that a good proportion of global *C. arabica* diversity was represented in our reference panel. Despite this, some Haitian samples did not cluster with any reference individuals.

## 3.2 Varietal assignment

Varietal assignment and characterization were carried out using a genetic structure analysis performed on both Haitian and reference samples with KASP SNP genotyping data. Cross-entropy for the sNMF structure analysis was lowest between K = 5–8. Increasing the number of clusters starting from K = 2 and up to K = 6 (Fig 1B) allowed for the identification of distinct varietal clusters labeled according to reference samples included therein. These were: a Typica-like (n = 263, including $n_H$ = 248 Haitian samples), Bourbon-like (n = 35, $n_H$ = 27), CR95/Catimor-like (n = 77, $n_H$ = 73), Kent/I-60-like (n = 9, $n_H$ = 7), and Ethiopian-like (n = 23, $n_H$ = 1) group, as well as a sixth group exclusively composed of Haitian samples (n = 52). The Ethiopian-like group was made up of collection accessions from *C. arabica*'s natural distribution range, with only one Haitian sample assigned to it.

Pairwise $F_{ST}$ values between the six genetic groups, considered as distinct varietal groups, showed them to be well-differentiated, whether calculated using only Haitian samples (Fig 2A) or including reference accessions. Overall, mean genetic distance between individuals in the same group was 0.016 ± 0.037 SD. The genetic group with the highest similarity between

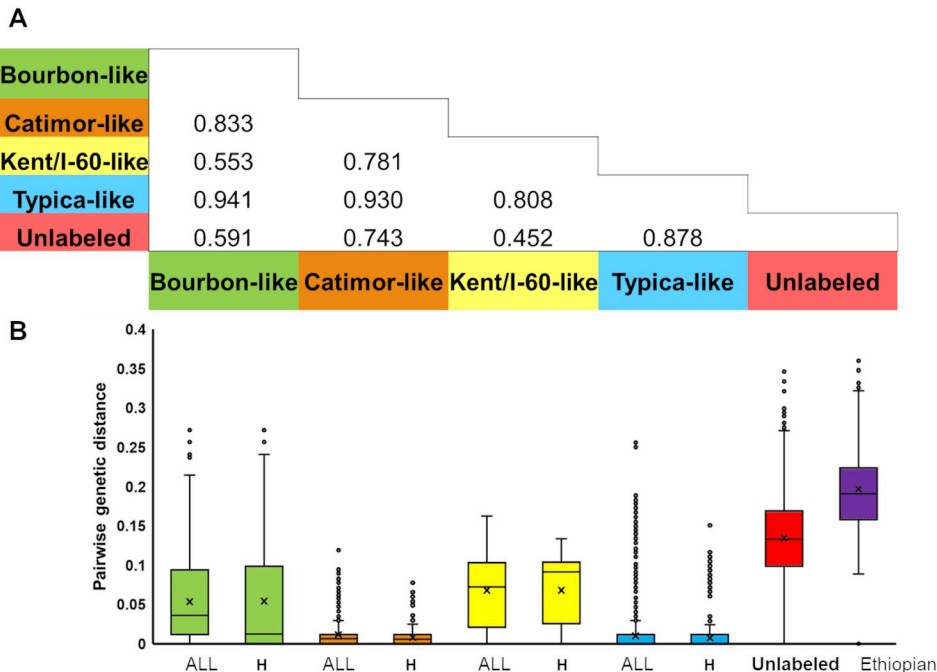

**Fig 2. Genetic differentiation between and within genetic groups. A.** Pairwise $F_{ST}$ between identified *C. arabica* genetic groups calculated on Haitian samples (N = 406) from data based on SNP genotyping data. All p-values <0.001 at 999 repetitions. **B.** Boxplot of pairwise simple matching genetic distances within genetic group for reference + Haitian (« All ») and Haitian-only (« H ») individuals. Colors correspond to those used for group labels in panel A. Only one Haitian sample was assigned to the Ethiopian-like group. The « Kent/I-60-like », « CR95/Catimor-like » and « Ethiopian-like » group names have been shortened for legibility.

individuals was Typica (0.010 ± 0.022, mean ± SD) and the genetic group with the highest mean genetic distance between individuals was the Ethiopian-like group (0.196 ± 0.057 SD) (Fig 2B).

Using the 80% membership threshold, 407 Haitian and 52 reference samples were assigned to a variety. 194 Haitian samples were considered to be admixed, with 128 of them having >40% membership from one genetic group, 20 samples from two groups (and always at least one of the two traditional lineages, Typica or Bourbon), and 46 samples never reaching 40% membership from any genetic group. 65 reference accessions were considered to be admixed.

The distribution on the PCA axes, and the clustering of individuals on the neighbor-joining dendrogram (Fig 1C) were consistent with results of the structure analysis. The first axis of the PCA primarily separated the Typica from the Bourbon groups, while the second axis primarily differentiated the CR-95 like and the "Unlabeled" groups (Fig 1A). The group of "Unlabeled" coffee trees were those which did not cluster with any reference sample in the PCA, confirming the absence of their representation in the CATIE international collection. The clustering of individuals on the neighbor-joining dendrogram was consistent with that of the structure analysis (Fig 1). The dendrogram showed Ethiopian accessions to be the most basal among *C. arabica* samples, and separated the other cultivated varieties in two main groups consistent with the historical Typica and Bourbon lineages and their derived varieties.

## 3.3 Comparison with HiPlex haplotype data

After filtering, 691 samples (including 94 *C. arabica*, three *C. canephora*, one *C. congensis*, and two *C. liberica* reference samples) and 225 multi-allelic haplotype markers were obtained from

the HiPlex amplicon sequencing data. Mean variant call error rate was 1.4% (min = 0.2%, max = 3.6%). Among *C. arabica* samples, we obtained SNP calls from KASP SNP genotyping data and HiPlex haplotype calls for 684 individuals (590 field and 94 reference samples). Genetic distance matrices per marker set were significantly correlated (p = 0.001, $R^2$ = 0.19). Likewise, there was a fairly linear relationship between the expected heterozygosity (gene diversity) calculated from both sets of markers at the farm (y = 1.01x+0.03, $R^2$ = 0.71) and municipality (y = 1.13x+0.01, $R^2$ = 0.99) levels (S1 Fig).

Population structure analysis for HiPlex haplotype data displayed low cross-entropy starting at K = 6, with a cluster of Ethiopian samples appearing at K = 7 (vs. K = 6 for the KASP SNP data). The accessions assigned to a genetic group with the KASP SNP data set were assigned to a corresponding HiPlex-based genetic group with a high frequency, although with a higher proportion for the large Typica groups (78.57%) than for the smaller Bourbon groups (46.88%), and the Kent/I-60-like group with only eight accessions could not be assigned with certainty with HiPlex haplotype data (S8 Table). The additional cluster detected from HiPlex haplotype data appears to be mostly made up of admixed individuals. The corresponding neighbor-joining dendrogram shows that relationships between the samples were generally conserved, although it split the Typica and Bourbon lines at a more basal position, before the Ethiopian-like group differentiation (S2 Fig).

### 3.4 Identification of Robusta coffee trees among Haitian samples

Several sampled individuals from one farm (N05) in northern Haiti, locally identified as a distinct variety, were suspected to belong to a different *Coffea* species based on visual aspect of the leaves and fruits and the lower genotyping success using Arabica-targeting markers. Using a subset of the 57 most complete markers across *Coffea* species, and thus recovering *C. congensis* and *C. liberica* samples which had been excluded from prior analyses, we calculated distance matrices and generated neighbor-joining dendrogram that suggested these individuals belonged to *C. canephora* (S3 Fig).

### 3.5 Farm composition

All genetic groups but the Ethiopian were identified in both departments (Fig 3). The Kent/I-60-like group was absent from two municipalities, Bahon and Grande Rivière du Nord, although Bahon only had one muti-varietal farm and Grande Rivière du Nord was only represented by a single (albeit multi-varietal) farm. The Bourbon-like group was also absent from the Grande Rivière du Nord farm. The composition of farms varied considerably. There were four monovarietal farms, which invariably consisted of the Typica genetic group, and had no admixed individuals. Two farms had individuals from the Typica and "Unlabeled" groups, and no admixed individuals. The 22 remaining farms had between two and five genetic groups, as well as admixed individuals of varying genetic backgrounds. Of the 148 admixed individuals with contribution from at least one genetic group at the >40% membership threshold, 68.2% co-occurred with plants from all contributing genetic groups. A further 4.7% had contribution from two genetic groups but co-occurred with plants from only one of those contributing groups. Finally, 40 out of the 148 admixed individuals (27.0%) were not cultivated on the same farms as any plant from the contributing genetic groups (Fig 4).

### 3.6 Local knowledge of variety mixtures

Five main vernacular category names were recorded during sampling, and all correspond to conventionally-used coffee varieties. The vernacular categories in Haitian creole were: *Tipika* (for "Typica", also called "Vieux café"), *Katoura* ("Caturra"), *Katimò* ("Catimor"), *Bougon* (for

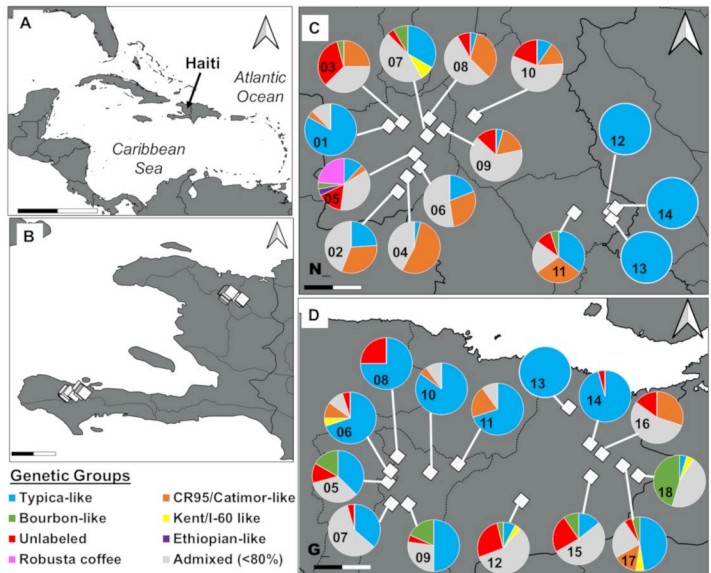

**Fig 3. Location and coffee genetic group composition of sampled farms in Haiti, based on SNP genotyping data.**
**A.** Geographic location of the Republic of Haiti in the Caribbean region. Scale bar ticks represent a distance of 250 km.
**B.** Study area in the Nord (N, northern) and Grande-Anse (G, southern) departments. Scale bar ticks represent 25 km.
**C.** Farm location and genetic group composition in the Nord department and **D.** in the Grande-Anse department.
Scale bar ticks in 3.C and 3.D. represent 2.5 km. Note: One farm (N05- *Bernice*) contained « Robusta coffee » (*C. canephora*) in addition to *C. arabica*. All other Haitian samples are *C. arabica*.

"Bourbon") and *Blou Monntenn* ("Blue Mountain"). A sixth name, *Kafe Brezil* ("Brazil Coffee"), was used to refer to the phenotypically distinct *C. canephora* individuals. 48 individuals could not be identified, and therefore were not assigned to a vernacular category.

Pearson's $\chi^2$ test revealed significant correlation between vernacular category and genetic group membership, whether considering hybrids as one category or sub-categorizing them ($p = 2.2e^{-16}$ in both cases). To illustrate the overlap between local knowledge and genetic identification, the percent representation of genetic groups in each vernacular category was plotted (Fig 5A). Despite the correlation found, vernacular categories were shared among samples belonging to various genetic groups, with some counterintuitive associations. For instance, "Blue Mountain" coffee generally refers to Jamaican Typica trees, which is consistent with the clustering of most reference Blue Mountain samples in the Typica genetic group. However, 31% of plants identified as *Blou Monntenn* were found to belong to the CR95/Catimor-like group. In fact, CR95/Catimor-like plants were more likely to be identified as *Blou Monntenn* (31.5%) than as *Katimò* (19.2%). Likewise, no plant assigned to the Bourbon-like genetic group was identified as *Bougon*, though 51.9% of them were identified as *Katoura*. The Caturra variety is a dwarf mutant of Bourbon coffee, and the structure analysis assigned Caturra reference samples to the Bourbon-like genetic group. 83.0% of samples from the Typica genetic group were identified as *Tipika*, and made up 64.1% of the latter. Of the 48 samples which had no clear vernacular category, 47.9% were admixed individuals and 27% were assigned to the Typica genetic group.

The number of varieties reported by farmers was not significantly correlated with the number of genetic groups ($p = 0.314$), nor was there a significant correlation between local perception of diversity and genetic group diversity ($p = 0.1199$) (Fig 5B). However, there was a

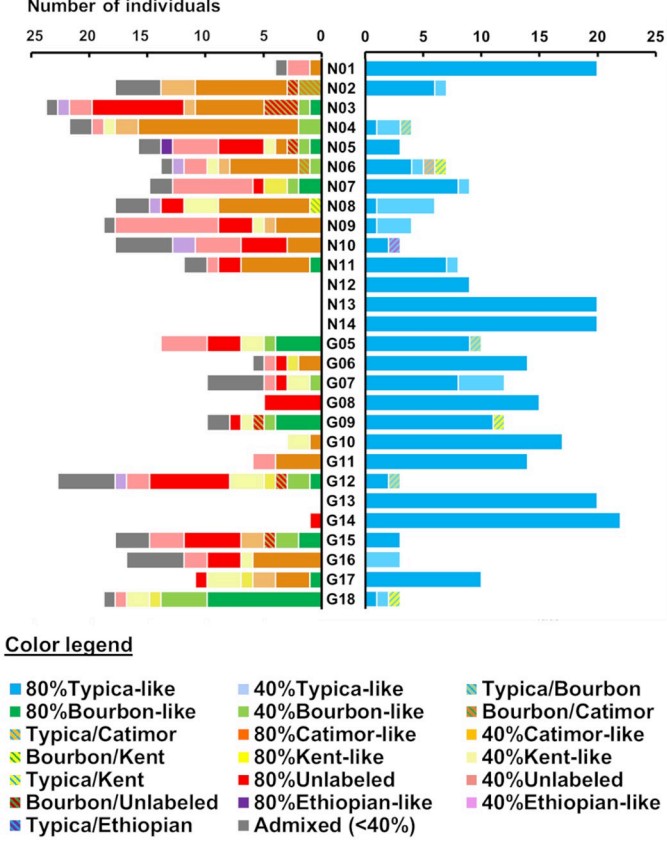

**Fig 4. Contribution of genetic groups and admixed individual to farm composition.** Number of *C. arabica* samples per farm with contribution from the historical Typica variety (right) and other genetic groups based on SNP genotyping data (left). Percentages in the legend refer to contribution thresholds. Hatched colors correspond to samples with >40% membership from two genetic groups. Admixed samples in grey never reached 40% membership from any genetic group. The « Kent/I-60-like » and « CR95/Catimor-like » group names have been shortened for legibility. Farms are noted Nx for the Nord (northern) and Gx for Grande-Anse (southern) departments, respectively.

significant association between farms' monovarietal or multi-varietal status as perceived locally and as determined by genetic group composition (p = 1.995e$^{-7}$).

## 4 Discussion

### 4.1 Two sets of markers for different genetic background signals

Cultivated Arabica coffee varieties originate mostly from two main lineages, Typica and Bourbon, as well as from crosses between them and with Ethiopian accession and the Timor Hybrid [29]. As such, we sought to distinguish between putative varieties using a set of markers (KASP SNP genotyping) designed for cultivated and wild accession diversity [39,53]. However, given the recorded historical introduction in Haiti of Typica and the early global spread of Bourbon, we also aimed to differentiate the two lineages using a multiplex amplicon sequencing assay (HiPlex) designed to discriminate the two varieties with variety-specific haplotypes [54].

Results acquired from both HiPlex haplotype and KASP SNP markers showed similar levels of genetic diversity across farms and municipalities, and both distances matrices were correlated. There were considerable similarities, but also notable differences between the structure

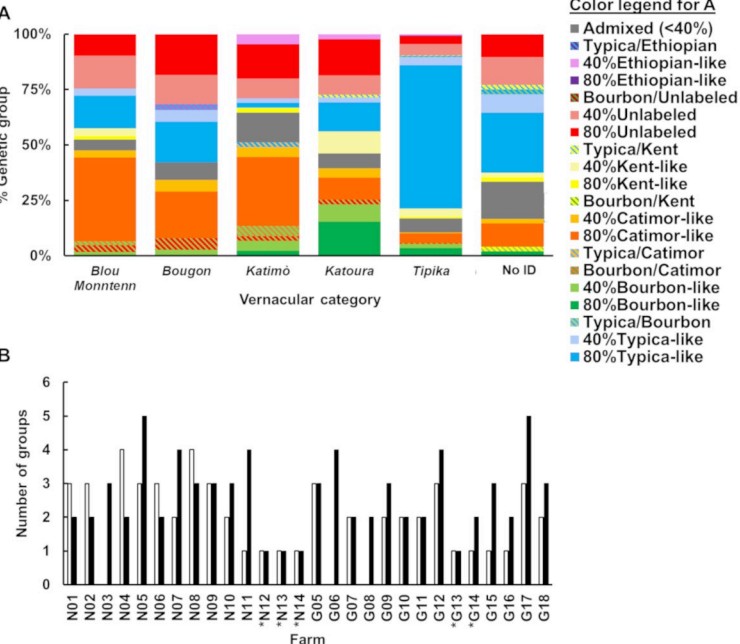

**Fig 5. Vernacular identification of Haitian *C. arabica* samples in relation to their assigned genetic groups based on SNP genotyping data. A.** Percent representation of each genetic group (and admixed individuals) in vernacular categories based on local knowledge. Color legend represent genetic composition, with hatched colors representing admixed individuals with >40% membership from two genetic groups. Admixed samples in grey do not reach 40% membership from any genetic group. The « Kent/I-60-like » and « CR95/Catimor-like » group names have been shortened for legibility. **B.** Number of vernacular varieties reported by farmers at time of sampling (open bars) and genetic groups identified by SNP genotyping (filled bars) in the Nord (Nx) and Grande-Anse (Gx) departments. Farms also had admixed individuals, except those marked with *. Where farmers were unsure of their farm's vernacular variety composition, no data is shown.

analyses (sNMF results): the most likely cluster number was K = 7 for the HiPlex haplotype data whereas it was K = 6 for the KASP SNP data, as these were the values at which Ethiopian-like genetic groups were identified. In both analyses, however, genetic groups were identified for major varieties, Typica-, Bourbon-, CR95/Catimor-like samples and the "Unlabeled" Haitian cluster with comparable level of assignation.

These overall differences may result from differences in the KASP and HiPlex marker assay design objectives. The KASP SNP marker set was developed to discriminate a broad panel of wild and commercial accessions while the HiPlex marker set was developed to maximize polymorphism between the two main cultivated varieties and identify crosses between them. In particular, by emphasizing the differences between Typica and Bourbon, these markers seem to introduce a level of ascertainment bias and erase the trace of the Ethiopian origin of these varieties (Ethiopian accessions are no longer basal on the tree). While we decided to focus on the KASP SNP genotyping data for varietal identification, the HiPlex haplotype assay may be useful in identifying further structuring in admixed individuals, when focusing on Typica x Bourbon hybrids, and are thus likely best used to infer intra-varietal structure.

Our study revealed considerable Arabica coffee diversity in Haitian CAFS. Five main varietal genetic groups were identified encompassing the two main cultivated Typica and Bourbon Arabica lineages. In fact, some farms had levels of gene diversity ($H_e$) only slightly lower than that measured across all reference *ex-situ* collection accessions, consistent with the PCA showing considerable overall between global Arabica diversity (as determined from our reference

accessions) and that present in Haitian samples. Farm composition was also quite variable: lower-diversity farms had up to two genetic (varietal) groups and little to no evidence of admixture, whereas high-diversity farms had two to five genetic groups as well as admixed individuals of varying genetic backgrounds.

## 4.2 Haitian farm diversity

Coffee genetic diversity showed no significant structure at the level of departments or municipalities suggesting that all varietal groups have been introduced or exchanged across both departments. Genetic variability was therefore highest within and between farms, with an overall lack of regional specificity or North-South divide. Nevertheless, there is a pattern of monovarietal farms being situated in more remote locations away from the main roads and larger cities. This is notably the case in the Bahon municipality in the North: the town is smaller and further away from the main northern city of Cap-Haitien, and the three monovarietal farms were in remote mountains accessible by foot or motorcycle, but not by car. The monovarietal farm in Grande-Anse was also the farthest from the main road. These farms have thus remained out of the reach for the many development projects that are involved in the introduction and spread of newer varieties.

## 4.3 Notes on varietal identification

Varietal identification by clustering of samples (as was done in the present study) can be complicated by the diversity of criteria used for defining a variety or accession [29,66,67]. Organoleptic properties, growth habits and other phenotypic traits, as well as geographic origins all factor in the concept of varieties in a way that could not be captured by our genetic methods. For instance, "Jamaican Blue Mountain" reference accessions were not genetically differentiated from other references in the Typica group, and "Caturra" and "Bourbon"-named reference accessions clustered together. The history of varietal naming in *C. arabica* is also complex. In the XX[th] century particularly, an abundance of new varieties and accessions were described from homozygote mutants and their outcrossing progeny [68]. Haarer (1923) gives the example of "Typica amarello" (*sic*), a yellow-fruited mutant of Typica, which was crossed with Bourbon to yield "Bourbon amarello", showing how nomenclature can obscure parentage. Indeed, in our study Typica Amarillo and Bourbon Amarillo references clustered together as admixed individuals. The matter is further complicated by recent genetic studies that have shed light on issues in mislabeling and maintenance of genetic integrity in coffee accessions worldwide [37,38]. Nevertheless, our methodology allows for the assignment of coffee plants to genetic groups that can be used to reconstruct pedigree relationships, infer cultivation history, predict agronomic performance, evaluate diversity and identify diversification processes in coffee farms.

## 4.4 Local perceptions of coffee varieties and diversity

Comparisons of the genetic and vernacular identification of sampled coffee plants revealed them to be correlated. Nevertheless, there were inconsistencies between the two categorizations. Montagnon [69] reported similar discrepancies between genetic and vernacular categorization of Arabica in traditional Yemeni farms, though by contrast to the latter study all vernacular categories recorded in Haiti corresponded to commercially-used varietal names. Overall, local knowledge is apt at distinguishing monovarietal stands from diversified ones, but the level of diversity is often underestimated, with some genetic groups and processes of admixture being overlooked. This is consistent with other studies comparing farmer-reported with genetically-determined varietal diversity [70,71].

Discrepancies may be in part due to farmers developing their own ways of categorizing plants based on practical concerns or phenotypic observations independent of plants' genetic parentage [72], even as the names used may be repurposed. For instance, plants from the Catimor-like genetic group had a similar likelihood of being assigned to the "*Katimò*" and "*Blou Monntenn*" vernacular categories. Speaking with farmers during sampling, it became apparent that *Blou Montenn* is often used to identify compact plants with shorter internodes. As generally understood, Blue Mountain coffee refers to Typica plants grown in the eponymous Jamaican mountains [73]. The attribution of this label to non-Typica plants in Haiti may originate from development projects mislabeling imported material, as one report referred to the implementation of varietal replacement in participating farms, whereby "the "Typica" coffee variety was substituted by the "Blue Mountain" variety, which was rust tolerant" [74]. The high proportion of admixed individuals could also be blurring the distinctions between varieties. Still, vernacular category names can provide insight to aid in interpreting genetic analyses. For instance, though neither market set distinguished Caturra from their Bourbon relatives, the presence of the *Katourra* vernacular category suggests that at least some of the genetic Bourbon-like Haitian samples could be Caturra. While one might expect this distinction to be easily resolvable through phenotypic observation, given Caturra's characteristic compact, "dwarf" growth form, the matter becomes considerably more complicated in the reality of Haitian fields. Indeed, stand ageing and lack of pruning has led to older plants acquiring tall, complex and extremely variable architectures that were quite different from those occurring in more diligently managed coffee stands.

### 4.5 Persistence of the historical Typica-like group

Plants in the Typica group were found in all farms except two. This group also made up the coffee stands in all four monovarietal farms, with two more being composed exclusively of plants from the Typica and Unlabeled group. Typica was thus the most widespread variety in the surveyed areas, which was expected, as this variety was the first to be introduced from a limited number of original clones and became widespread both during and after the colonial era. This variety is the traditional "*Vieux café*" from Haiti, and our analysis confirms that despite the genetic mixing taking place in Haitian CAFS, it has retained its genetic integrity both in northern and southern Haiti. Typica have very good potential for quality but display low yields and high susceptibility to disease [73].

The Typica cluster was found to have the least diversity, which is consistent with the strong population bottleneck that resulted from the introduction of this lineage to the Neotropics, the first instance of coffee being grown in the Americas. This group had the most overlap between genetic groups and vernacular categories, and local farmers were more likely to describe plants from other genetic groups as "*Tipika*" than they were to ascribe plants from the Typica-like genetic group to other vernacular categories. This suggests a high degree of familiarity with the *Vieux Café*, and that Typica is still seen as the "quintessential" Haitian coffee variety. Anecdotally, during sampling, most farmers described their fields as "*Vieux Café / Tipika*" stands, supplemented in many cases by other varieties acquired more recently from cooperatives, nurseries or development projects.

### 4.6 Varietal groups beyond Typica established in Haitian CAFS

**4.6.1 Bourbon-like group.** Individuals assigned to the Bourbon-like genetic group were found on ten farms. Notably, none of the vernacular "*Bougon*" were assigned to this group, though many "*Katoura*" were. We could not determine whether the Bourbon-like group came from historical or more recent introductions. The Bourbon variety has been cultivated in the

Neotropics since the XVIII[th] [68] or mid-XIX[th] century [73], though we found no record of early introduction to Haiti. Seeds from varieties named Bourbon (H33) and Caturra (T2308) were imported to Haiti in the 1970s from Turrialba, Costa-Rica [51]. Caturra was noted to have an especially vigorous growth, with monovarietal fields being established in Southeast Haiti. Despite their genetic closeness, Bourbon has better cup quality potential and lower nutrition requirements but also lower yields than Caturra, making further investigation into the Haitian plants' identity important. Both are susceptible to coffee leaf rust and other diseases.

**4.6.2 CR95/Catimor-like group.** Over half of the surveyed farms contained individuals that clustered with CR95 reference accessions: ten in the North and five in Grande-Anse. CR95 is part of the coffee leaf rust tolerant Catimor group originating from a cross between Caturra and the Timor Hybrid, though loss of resistance has been reported. These are high-yielding plants with lower quality potential and high nutrition requirements [73]. Trials of the then-experimental Catimor variety (T5159) were conducted in Haiti as early as 1977 (Ester, 1978), and it was widely disseminated in Haiti by various agricultural development projects in the 1990s [52]. As previously discussed, Catimors may have been disseminated under other names through distribution networks.

**4.6.3 Kent/I-60-like group.** Only five farms (just one in the North) were found to contain Kent/I-60-like trees, though almost half of them (13 farms) had plants with <40% membership from Kent/I-60-like ancestors. The Kent variety originated in 1911 on an eponymous estate in Mysore, India. At the time, it boasted (now-lost) resistance to coffee leaf rust, leading to its widespread cultivation in India and East Africa [68]. Though to our knowledge no record exists of its introduction in Haiti, Haarer, writing in 1923 [68] makes clear that it enjoyed a great reputation, deeming it the best choice should seed be imported from another country. The relative rarity of Kent/I-60-like Haitian individuals and the widespread presence of admixed individuals is consistent with an old introduction of this variety.

**4.6.4 Unlabeled group.** Population structure analyses identified a varietal cluster containing no reference individuals, making its identification difficult. It was present in 17 of the sampled farms. The closest reference sample was a Sarchimor (Villa Sarchi Bourbon mutant x Timor Hybrid, WCR) with 50.0% likelihood of contribution from this group. The "Unlabeled" plants could potentially belong to Timor Hybrid-introgressed Colombian varieties Tabi and Castillo, which are known to have been used in trials in both departments starting in 2013 [52]. However, no farmer reported growing them, and they are absent from the CATIE international collection. This group's widespread presence on sampled farms suggest that they are readily spread through distribution networks, though seemingly without a specific vernacular name.

**4.6.5 Inter-varietal individuals.** Admixed individuals between two genetic groups were at least twice more likely to be found in farms where at least one, and more often two of their contributing parental varieties were represented, suggesting that at least some were the result of genetic mixing within their farm. Renewal of coffee stands occurs primarily through germination of the seed bank, though some farmers also source their trees from nurseries or development programs [48,52]. While *C. arabica* is considered to be mainly autogamous, its rate of allogamy has been estimated at >10% [75] and significantly more in some cases [76], allowing gene flow and recombination between individuals [37]. Furthermore, pollinators can increase the seed set of *C. arabica* [77], and are more numerous in agrosystems with more diversity and lower use of chemical inputs [78], potentially favoring outcrossing. The within-farm production of admixed individuals could therefore result from *in-situ* outcrossing and subsequent recruitment from the seed bank. Very little scientific attention has been paid to the varietal mixtures and genetic mixing in *C. arabica* fields outside of the Ethiopian accessions in the

species' geographic area of origin [76,79]. While our analyses agreed with previous reports of considerable diversity in Ethiopian accessions, our results highlight the potential role of non-intensive, no-input CAFS in generating diversity from cultivated accessions under appropriate conditions. As Haitian CAFS are recovering from the severe 2012 rust epidemic, which impacted much of the Neotropical region [52,80], natural selection may have favored admixed individuals with greater resistance to the disease. This could explain the high proportion of admixed individuals with contribution from the CR95/Catimor-like group and perhaps also the "Unlabeled" group.

Two alternate or complementary explanations for inter-varietal individuals could be proposed. Firstly, there are reports of the Mondo Novo variety being introduced to Haiti, though it is said to have failed to establish in the fields [50]. Survival of this cross between the Typica and Bourbon varieties may account for some of the admixed individuals in Haitian farms. As such individuals were present in all diversified farms, their production may also be taking place in nurseries or in certain specific farms from which they are spread through networks of seed and seedling exchange. These processes might be strengthened by aforementioned inaccuracies in local assessment of farm varietal composition.

Six admixed samples were identified as having >40% membership from the Ethiopian-like group (in addition to a single individual from Haiti assigned to that genetic group). One possible source is the Geisha variety, which is known for its high cup quality potential. It was reportedly used in trials in the 1970s [51], though mentions of it are all but absent from subsequent sources. The Geisha sample in our reference panel was itself considered admixed, with 46.8% contribution from the Ethiopian-like genetic group.

**4.6.6 Robusta coffee: An unreported crop in Haitian CAFS.** One farm in the North was also found to contain *C. canephora*, a species whose presence in Haiti had not been previously recorded to our knowledge. *C. canephora*, or "Robusta coffee", is the second most economically important *Coffea* species [23]. In Haiti, it was identified as *Kafé Brezil* ("Brazil Coffee"), potentially pointing to an introduction from this country where it is widely cultivated. A report from the South of Haiti also makes mention of "Brazil Coffee", referring to it as a new, rust-resistant but less productive "variety" [50] rather than another species altogether. The cultivation of Robusta is more recent, and its expansion since the end of the XIX[th] century is linked, among others, to the increase in demand for coffee worldwide, the susceptibility of *C. arabica* to coffee leaf rust, despite its lower potential cup quality [38]. A recent trend towards increased cultivation of this species in Latin American countries, where only Arabica was previously grown, has also been noted [81]. This is due to Robusta's attractive qualities of greater rust resistance but also to the fact that it is more resilient to the effects of climate change. Cultivation of *C. canephora* appears to be relatively rare in Haiti at present.

## 4.7 Maintenance and generation of diversity in Haitian CAFS: Specificity, implications and future directions

Harvey et al. [81] noted a trend of widespread replacement of thousands of hectares of traditional varieties in Latin America, by disease-resistant (and especially rust-resistant) cultivars such as Catimors and Sarchimors, derived from Timor Hybrids. A rise in the intensification of cultivation was also observed, albeit with a concurrent increase in the area of coffee cultivated under voluntary sustainability standards. By contrast, disease-resistant varieties have been introduced to Haitian CAFS in the North and Grande-Anse departments, but have not replaced traditional varieties and indeed appear to have mixed with them under consistently low-intensity management. This complicates varietal identification and variety-specific

management in the field, but also allowed for the generation of considerable genetic diversity, as revealed by our analyses.

In addition to resistant cultivars, F1 hybrids bred for heterosis from traditional varieties and more distant Ethiopian accessions have been developed [82]. These hybrids are increasingly promoted as a response to concerns over quality, productivity, disease resistance, and the need for ecologically sustainable shade-adapted coffee plants. In a study of acceptance by Central American farmers, these hybrids, when available, were found to be attractive [83]. However, they come with the caveat that the vast majority of them do not breed true, and must therefore be propagated clonally or purchased. This is important given the economic context and management system in Haitian CAFS. Indeed, any prospect of importing such hybrids must consider that the dynamic process of genetic mixing and seedling recruitment taking place in Haitian CAFS might lead to unexpected outcomes.

The considerable Arabica genetic diversity found in Haitian CAFS may contribute to the coffee sector's continued survival despite the considerable challenges it faces by providing potential for adaptation to biotic and abiotic stressors. Furthermore, given the process of genetic mixing among genetic groups in Haitian CAFS, this environmental pressure could promote processes of natural selection and adaptation. This likelihood is increased by the presence in various farms of rust-resistant cultivars such as Catimor (and possibly the "Unlabeled" group). In addition, this process could be supplemented by human-mediated selection of select, agronomically attractive plants to conserve, propagate and exchange [84]. Measures of the production potential and pest and disease burden at the plant and farm level in Haitian CAFS are needed to test these hypotheses. In addition, admixed individuals should be investigated for their potential agronomic performances and organoleptic qualities in the search for genotypes of interest. Finally, propagation networks should also be studied to better understand the processes of genetic diversification in Haitian CAFS.

## 4.8 Leveraging Haitian coffee diversity

Increased understanding of the diversity and varietal composition identification of Haitian CAFS can better support farmers' decision-making and inform field management, as varieties differ in their agronomic characteristics and requirements [73]. For instance, farmers interested in accessing specialty coffee markets may choose to focus on varieties with higher cup quality or historical-cultural value such as the Typica *Vieux Café* despite lower yields and higher pest and disease pressure. By contrast, those interested in producing dry-processed coffee beans for local markets may instead value higher-yielding genotypes that are more resistant to disease such as Catimor, and those for whom coffee is not a priority crop in their diverse CAFS may focus on plants tolerant of neglect. Furthermore, given that potential coffee cup quality is partially genetically determined [29,85,86], increased accuracy in the knowledge of farms' standing diversity and varietal makeup can help inform the commercialization of harvested coffee beans by better describing the varietal blends produced at a farm or regional level. Finally, this knowledge can allow CAFS to function as *in-situ* conservatories of the country's coffee germplasm resource [36], as well as sources for their propagation and dissemination. This can occasion a shift of farmers' perceived role from mere purveyors of economic product to active participants in the management of the country's genetic resources.

## 4.9 Conclusion

Our results indicate that Haitian coffee farms act not only as repositories of heritage varieties, but as generators of new genotypic combinations. This study is among the first to study *in-situ* coffee variety mixtures in general and to characterize that of Haitian coffee farms in particular.

The Haitian coffee sector would benefit from further studies, including phenotypic characterization which may help identify genetic material that is particularly well adapted to the local ecological and agronomic conditions. Future directions may also include the establishment of a Haitian germplasm conservation center. This genetic heritage contributes to the potential for renewal of the Haitian coffee sector which may in turn help maintain the provision of crucial ecosystem services by agroforestry systems. Shade-grown coffee could thus help safeguard Haitian biodiversity and improve community resilience and food security.

## Supporting information

**S1 Table. Name, location and number of samples for each study site.**
(DOCX)

**S2 Table. List of sample accessions used as references in the study.** List of reference samples by code and accession name, country and region of origin (when this information is known to the authors), and name of the holding institution whose collection the sample was acquired from.
(DOCX)

**S3 Table. Haitian *C. arabica* diversity statistics calculated on SNP genotyping data for the sampled departments of Nord (N) and Grande-Anse (G), and both combined (N+G): Sample size, observed heterozygosity ($H_o$), expected heterozygosity (= gene diversity, $H_e$), Fixation index (as $F_{IS}$) and percent marker polymorphism (% P).**
(DOCX)

**S4 Table. Haitian *C. arabica* diversity statistics calculated on SNP genotyping data for the sampled municipalities (*communes*) in the Nord (N) and Grande-Anse (GA) departments: Sample size (N), observed heterozygosity ($H_o$), expected heterozygosity (= gene diversity, $H_e$), Fixation index (as $F_{IS}$) and percent marker polymorphism (% P).**
(DOCX)

**S5 Table. Haitian *C. arabica* diversity statistics for the sampled farms in the Nord (N) and Grande-Anse (G) departments, based on SNP genotyping data: Sample size (N), observed heterozygosity ($H_o$), expected heterozygosity (= gene diversity, $H_e$), Fixation index (as $F_{IS}$) and percent marker polymorphism (% P).**
(DOCX)

**S6 Table. Pairwise $F_{ST}$ values between Haitian *C. arabica* sampled in municipalities (*communes*, M) of two departments (D), calculated from KASP SNP genotyping data.**
(DOCX)

**S7 Table. Pairwise $F_{ST}$ values between Haitian *C. arabica* sampled in farms (F) from five municipalities (M) and two departments (D), calculated from KASP SNP genotyping data.**
(DOCX)

**S8 Table. Correspondence between genetic groups defined on KASP SNP genotyping or HiPlex haplotype data.**
(DOCX)

**S1 Fig. Expected heterozygosity ($H_e$) calculated on HiPlex haplotypes *versus* KASP SNP genotyping.**
(DOCX)

**S2 Fig. Unweighted neighbor-joining dendrogram of Haitian coffee samples and reference samples from collections ("C") based on HiPlex haplotypes.**
(DOCX)

**S3 Fig. Unweighted neighbor-joining dendrogram of Haitian Robusta coffee trees and reference samples.** Supplementary Note: HiPlex read processing bioinformatics pipeline.
(DOCX)

## Acknowledgments

First and most importantly, the authors wish to thank the Haitian farmers who have welcomed us into their fields and given access to them. We would also like to thank the AVSF and PITAG teams who provided logistical support throughout the sampling process, especially Rochelin Pierre-Louis, Franndy Mulatre and Agronome Magency, as well as the Haitian Ministry of Agriculture, Natural Resources and Rural Development (MARNDR) who pilots the PITAG project and provided the permits for exporting genetic material. We also thank Chifumi Nagai and Ming-Li Wang for providing the HARC reference samples and passport information. We also express our sincere gratitude to Vincent Soulé and Kurt Lamour for their assistance with molecular lab methods, to Adeline Barnaud, Julie Orjuela and Yves Bawin for their scientific insights, and to Louis Champion and Killian Laurent for their software help. We wish to acknowledge the ISO 9001 certified IRD i-Trop HPC (member of the South Green Platform) at IRD Montpellier for providing HPC resources that have contributed to the research results reported within this paper. URL: https://bioinfo.ird.fr/- http://www.southgreen.fr".

## Author Contributions

**Conceptualization:** Claude Patrick Millet, Clémentine Allinne, Pierre Marraccini, Tom Ruttink, Dapeng Zhang, Valérie Poncet.

**Data curation:** Claude Patrick Millet, Dapeng Zhang, Valérie Poncet.

**Formal analysis:** Claude Patrick Millet, Lauren Verleysen, Valérie Poncet.

**Funding acquisition:** Clémentine Allinne, Wesly Jeune.

**Investigation:** Claude Patrick Millet, Clémentine Allinne, Valérie Poncet.

**Methodology:** Claude Patrick Millet, Clémentine Allinne, Tram Vi, Lauren Verleysen, Marie Couderc, Dapeng Zhang, Christine Tranchant-Dubreuil, Valérie Poncet.

**Project administration:** Clémentine Allinne, William Solano Sanchéz, Wesly Jeune.

**Resources:** Claude Patrick Millet, Clémentine Allinne, William Solano Sanchéz, Wesly Jeune.

**Software:** Claude Patrick Millet, Tram Vi, Lauren Verleysen.

**Supervision:** Clémentine Allinne, Pierre Marraccini, Marie Couderc, Tom Ruttink, Christine Tranchant-Dubreuil, Valérie Poncet.

**Validation:** Claude Patrick Millet, Clémentine Allinne, Valérie Poncet.

**Visualization:** Claude Patrick Millet.

**Writing – original draft:** Claude Patrick Millet.

**Writing – review & editing:** Claude Patrick Millet, Clémentine Allinne, Lauren Verleysen, Tom Ruttink, Dapeng Zhang, Wesly Jeune, Valérie Poncet.

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
