## [Decision Letter · Decision Letter 0]

23 Oct 2023

PONE-D-23-28150Haitian Coffee agroforestry systems harbor considerable, dynamic and under-reported variety mixtures and genetic diversityPLOS ONE

Dear Dr. Millet,

Thank you for submitting your manuscript to PLOS ONE. After careful consideration, we feel that it has merit but does not fully meet PLOS ONE’s publication criteria as it currently stands. Therefore, we invite you to submit a revised version of the manuscript that addresses the points raised during the review process.

We look forward to receiving your revised manuscript.

Kind regards,

Awatif Abid Al-Judaibi, PhD

Academic Editor

PLOS ONE

Journal Requirements:

"C.P.M. was funded by PhD grants from the French Embassy in Haiti and the ARTS program (IRD).This work was supported by the Agricultural and Agroforestry Technological Innovation Program (HA-L1107, HA-G1038) funded by the IDB, GAFSP, IFAD and Haitian government. L.V. was supported by Research Foundation-Flanders (FWO; G090719N)." 

5. We note that Figure 4 in your submission contain [map/satellite] images which may be copyrighted. All PLOS content is published under the Creative Commons Attribution License (CC BY 4.0), which means that the manuscript, images, and Supporting Information files will be freely available online, and any third party is permitted to access, download, copy, distribute, and use these materials in any way, even commercially, with proper attribution. For these reasons, we cannot publish previously copyrighted maps or satellite images created using proprietary data, such as Google software (Google Maps, Street View, and Earth). For more information, see our copyright guidelines: http://journals.plos.org/plosone/s/licenses-and-copyright.

a. You may seek permission from the original copyright holder of Figure 4 to publish the content specifically under the CC BY 4.0 license.  

6. We notice that your supplementary tables are included in the manuscript file. Please remove them and upload them with the file type 'Supporting Information'. Please ensure that each Supporting Information file has a legend listed in the manuscript after the references list.

Reviewers' comments:

Reviewer's Responses to Questions

**Comments to the Author**

1. Is the manuscript technically sound, and do the data support the conclusions?

Reviewer #1: Yes

Reviewer #2: Yes

2. Has the statistical analysis been performed appropriately and rigorously? 

Reviewer #1: Yes

Reviewer #2: No

3. Have the authors made all data underlying the findings in their manuscript fully available?

Reviewer #1: No

Reviewer #2: Yes

4. Is the manuscript presented in an intelligible fashion and written in standard English?

Reviewer #1: Yes

Reviewer #2: Yes

5. Review Comments to the Author

Reviewer #1: The authors studied the genetic diversity of Coffee arabica plants found at two discreet locations with historical importance found in Northern and Southern Haiti. This study revealed higher than expected genetic diversity and the presence of clades (e.g., Typica, Burbon, and Catimor) at the farm level; suggesting the crops of individual growers represent diverse coffee. Although additional work will need to be done to identify if these results are representative. The authors link “local knowledge” with clades and identify a -relatively-robust degree of concurrence, particularly for discriminating from monoculture. The authors identify coffee farms as repositories of varietals and sources of genetic materials, an important first step to cataloging plants that may benefit Haitian Coffee Agroforestry Systems in face of global climate changes and variable growing conditions.

While I acknowledge there are regulations that may limit the sharing of data, I would encourage uploading of data to public repositories (where permissible), the identification of accession# for genetic/genomic information available from reference materials (when available), and/or use public repositories like GitHub or protocols.io. This would greatly improve reproducibility and reanalysis of sequencing data in future studies.

Lines 125-128 – SNPs are an effective, validated method to characterize diversity and used to discriminate across several systems (prokaryotes and eukaryotes). When combined with other data sources can be used to reveal structural variation. However, they usually cannot, de novo, reveal structure (e.g., indels and CNVs) in most systems. Please clarify if the intent was structural genomic variation for C. arabica. Please also include additional references as Bawin was not accessible to me (login required) and Zhang does not provide sufficient detail.

A table or additional accounting in M&M as to the number of specimens from each farm/locality as well as the number of specimens from each reference set (CATIE, HARC, IRD) would aid a reader with a wholistic understanding of the total number (representants) and source of the diversity for both the HiPlex and KASP data e.g., is the only C. congensis from IRD.

It appears the figure legends and figure numbers do not match for Figure 3 and Figure 4.

Minor – In general, the manuscript is well written however, I encourage the authors to review the grammar/spelling to ensure it is of high quality and conveys the impact of their work clearly. For example, line 32, “as well as”, line 91 spacing before bracket, line 160 is missing a period, line 231 “amplicon”, line 254 “R” is a statistical software and should be capitalized, to identify century both Roman numerals and numbers are used, etc.

In line 113 can the definition (e.g., rust tolerance) of “elite” be provided as it seems to be used for various intents in the literature? Also, is the possessive form (apostrophe) intended?

In line 141 can the authors expand on what, “high-potential” refers to?

In line 200 can the number of specimens sequenced from HARC be identified?

Reviewer #2: The study is well organized

Title: needs clarification Suggest adding remove under-reported with other sentence

There is no need to use keywords that are already present in the title.

in line 172 Selection of study sites not clear please suggst screening the location for study

Deep discussions and related recent studies must be included in the result and discussion part.

6. PLOS authors have the option to publish the peer review history of their article (what does this mean?). If published, this will include your full peer review and any attached files.

Reviewer #1: No

Reviewer #2: No

---

## [Author Response · Author response to Decision Letter 0]

30 Nov 2023

Response to comments from the Editor

Journal Requirements:

Response: I apologize for this oversight and will ensure the proper naming format is followed.

"C.P.M. was funded by PhD grants from the French Embassy in Haiti and the ARTS program (IRD). This work was supported by the Agricultural and Agroforestry Technological Innovation Program (HA-L1107, HA-G1038) funded by the IDB, GAFSP, IFAD and Haitian government. L.V. was supported by Research Foundation-Flanders (FWO; G090719N)." 

Please state what role the funders took in the study. If the funders had no role, please state: ""The funders had no role in study design, data collection and analysis, decision to publish, or preparation of the manuscript."" If this statement is not correct you must amend it as needed. Please include this amended Role of Funder statement in your cover letter; we will change the online submission form on your behalf.

Response: I have added your statement to the end of ours, as it is indeed accurate to our situation. We have also added an additional source of support for a co-author.

Response: In order to comply with Nagoya protocol, we signed an MTA with the Haitian Ministry of Agriculture, in which they allow for reuse of the produced data, but require us to inform them. Therefore, we published it to the IRD dataverse repository, with the DOI https://doi.org/10.23708/T6YZML, but will still require that user contact the IRD in order to download them. This will allow us to comply with the Ministry’s request.

Response: We have added an ethics statement as applicable to our study in section 2.1 “Selection of Study Sites”. Namely, we state that we explained to farmers the purpose and aims of our study, acquired signed Prior Informed Consent forms from each of them, and that we also obtained export permits from the Haitian Ministry of Agriculture. 

We've also added a mention of Material transfer agreements (MTAs) and export permits, but we wonder if it is really necessary to mention it in this section. If not, please feel free to remove it.

5. We note that Figure 4 in your submission contain [map/satellite] images which may be copyrighted. All PLOS content is published under the Creative Commons Attribution License (CC BY 4.0), which means that the manuscript, images, and Supporting Information files will be freely available online, and any third party is permitted to access, download, copy, distribute, and use these materials in any way, even commercially, with proper attribution. For these reasons, we cannot publish previously copyrighted maps or satellite images created using proprietary data, such as Google software (Google Maps, Street View, and Earth). For more information, see our copyright guidelines: http://journals.plos.org/plosone/s/licenses-and-copyright.

a. You may seek permission from the original copyright holder of Figure 4 to publish the content specifically under the CC BY 4.0 license. We recommend that you contact the original copyright holder with the Content Permission Form (http://journals.plos.org/plosone/s/file?id=7c09/content-permission-form.pdf) and the following text: “I request permission for the open-access journal PLOS ONE to publish XXX under the Creative Commons Attribution License (CCAL) CC BY 4.0 (http://creativecommons.org/licenses/by/4.0/). Please be aware that this license allows unrestricted use and distribution, even commercially, by third parties. Please reply and provide explicit written permission to publish XXX under a CC BY license and complete the attached form.”

 Please upload the completed Content Permission Form or other proof of granted permissions as an ""Other"" file with your submission. In the figure caption of the copyrighted figure, please include the following text: “Reprinted from [ref] under a CC BY license, with permission from [name of publisher], original copyright [original copyright year].”

Response: As the figure in question is an important illustration of our results, we did not remove it. Instead, and in order to avoid any and all issues, we remade it, simply using shapefiles from Natural Earth, as suggested in your e-mail. We thank you for providing us with recommendations for sourcing freely useable material.

6. We notice that your supplementary tables are included in the manuscript file. Please remove them and upload them with the file type 'Supporting Information'. Please ensure that each Supporting Information file has a legend listed in the manuscript after the references list.

Response: We have fixed this error and provided the list of Supporting information legends.

Response: We have indeed checked our references and, have made the following changes to our cited references list:

(line) I added a citation (Engelmann et al., 2007) to the claim of CATIE collection’s international importance.

(line 129-130 in “track changes” manuscript): At reviewer #1’s request, we have included more appropriate references to studies illustrating the statement we made regarding the benefits of marker genotyping studies for exploring coffee genetic diversity. I added work by Scalabrin et al. (2020), Sousa et al. (2017), Zewdie et al. (2022) and Zhou et al. (2016).

(lines 720-740 in “track changes” manuscript): in section 4.7, I added two paragraphs discussing Haitian diversity in relation to recent/emerging trends in coffee cultivation, for which I also added a few references. Namely, I cited papers from Harvey et al (2021), Turreira-Garcia (2022) and Van der Vossen et al (2015).

We hope that the steps taken to address your various comments and concerns will prove satisfactory, and we once again thank you for your time, attention to- and comments on our study.

 

Response to comments from Reviewers

5. Review Comments to the Author

Reviewer #1

Reviewer #1: The authors studied the genetic diversity of Coffee arabica plants found at two discreet locations with historical importance found in Northern and Southern Haiti. This study revealed higher than expected genetic diversity and the presence of clades (e.g., Typica, Burbon, and Catimor) at the farm level; suggesting the crops of individual growers represent diverse coffee. Although additional work will need to be done to identify if these results are representative. The authors link “local knowledge” with clades and identify a -relatively-robust degree of concurrence, particularly for discriminating from monoculture. The authors identify coffee farms as repositories of varietals and sources of genetic materials, an important first step to cataloging plants that may benefit Haitian Coffee Agroforestry Systems in face of global climate changes and variable growing conditions.

(1.1) While I acknowledge there are regulations that may limit the sharing of data, I would encourage uploading of data to public repositories (where permissible), the identification of accession# for genetic/genomic information available from reference materials (when available), and/or use public repositories like GitHub or protocols.io. This would greatly improve reproducibility and reanalysis of sequencing data in future studies.

See Response to editor’s comment on the same topic.

(1.2) Lines 125-128 – SNPs are an effective, validated method to characterize diversity and used to discriminate across several systems (prokaryotes and eukaryotes). When combined with other data sources can be used to reveal structural variation. However, they usually cannot, de novo, reveal structure (e.g., indels and CNVs) in most systems. Please clarify if the intent was structural genomic variation for C. arabica. Please also include additional references as Bawin was not accessible to me (login required) and Zhang does not provide sufficient detail.

Response: In the highlighted lines, we were referring to population structure as understood through the lens of population genetics, rather than referring to structural variants in the genome. You are certainly right in pointing out that this can easily lead to confusion, and we have therefore reformulated the sentence in question in order to make clearer our statement. We have also provided other references to illustrate our point, as requested. L 130-133

(1.3) A table or additional accounting in M&M as to the number of specimens from each farm/locality as well as the number of specimens from each reference set (CATIE, HARC, IRD) would aid a reader with a wholistic understanding of the total number (representants) and source of the diversity for both the HiPlex and KASP data e.g., is the only C. congensis from IRD.

Response: We have included an additional Supplementary table detailing the reference accessions used in our study. We have included their origin, but also stated whether they were part of the panel used by Zhang et al (2021) to select the core set of SNPs used in the KASP targeted genotyping study. In addition, we have stated the genetic group each accession was assigned to using the 80% threshold in population structure analyses using both the KASP genotyping and the HiPlex Haplotype data, in order to both provide an additional layer of results and also show which accessions were represented in only one, or in both sets of data. 

(1.4) It appears the figure legends and figure numbers do not match for Figure 3 and Figure 4.

Response: This is an oversight which we have rectified.

(1.5) Minor – In general, the manuscript is well written however, I encourage the authors to review the grammar/spelling to ensure it is of high quality and conveys the impact of their work clearly. For example, line 32, “as well as”, line 91 spacing before bracket, line 160 is missing a period, line 231 “amplicon”, line 254 “R” is a statistical software and should be capitalized, to identify century both Roman numerals and numbers are used, etc.

Response: We thank you for both your positive comments on the manuscript overall and for taking the time and care to point out specific errors we missed during our edits. We have tried to fix them, as well as a few minor others we found upon re-reading. 

(1.6) In line 113 can the definition (e.g., rust tolerance) of “elite” be provided as it seems to be used for various intents in the literature? Also, is the possessive form (apostrophe) intended?

Response: The use of possessive single apostrophes was a mistake we have fixed. Regarding the use of “elite”, we were referring to the modern clonally propagated F1 hybrids that seek to combine qualitative traits, productivity, and resistant to leaf rust and berry disease. We removed the term and reformulated this sentence to simply mention that the CATIE collection holds breeds as well as hybrids bred for these various traits. L. 115-117

In line 141 can the authors expand on what, “high-potential” refers to?

Response: The term “high-potential” referred to Haitian coffee’s ability to be attractive to specialty markers, but we acknowledge that this term, being polysemantic and somewhat vague, can lead to confusion, especially given that Haiti does not compete significantly on global markets. We have modified the sentence to make it clearer. L. 147-148

(1.7) In line 200 can the number of specimens sequenced from HARC be identified?

Response: We have specified this number, as well as that of reference specimens of different species sampled from the IRD greenhouse. L. 211-217 

Reviewer #2

Reviewer #2: The study is well organized

(2.1) Title: needs clarification Suggest adding remove under-reported with other sentence

Response: We have changed the title from “Haitian Coffee agroforestry systems harbor considerable, dynamic and under-reported variety mixtures and genetic diversity” to “Haitian coffee agroforestry systems harbor complex Arabica variety mixtures and under-recognized genetic diversity” which we believe is clearer

(2.2) There is no need to use keywords that are already present in the title.

Response: We had chosen keywords that best described our study to make it findable in a keyword-based search. 

(2.3) in line 172 Selection of study sites not clear please suggest screening the location for study

Response: We are not sure what is meant by this comment. Still, we have tried to provide additional information in section 2.1 “Selection of Study Sites”. We have also included an additional supplementary table providing the names, geographic location and number of collected samples for each of the study sites. In doing so, we hope that we have adequately addressed your concerns. 

(2.4) Deep discussions and related recent studies must be included in the result and discussion part.

Response: Again, we are not certain what this comment means. There is little in the literature on Haitian coffee, and since much of our paper is devoted to discussing the genetic results in light of available reports, we tried to use the best available sources. However, we thought that perhaps this comment could be suggesting that we place our findings in the broader context of recent developments in coffee science and/or cultivation. We have therefore added a few paragraphs doing just that. We hope that we did not misunderstand, and that our response was satisfactory.

---

## [Decision Letter · Decision Letter 1]

12 Feb 2024

Haitian coffee agroforestry systems harbor complex Arabica variety mixtures and under-recognized genetic diversity

PONE-D-23-28150R1

Dear Dr. Claude Patrick Millet,

We’re pleased to inform you that your manuscript has been judged scientifically suitable for publication and will be formally accepted for publication once it meets all outstanding technical requirements.

Kind regards,

Awatif Abid Al-Judaibi, PhD

Academic Editor

PLOS ONE

Reviewers' comments:

Reviewer's Responses to Questions

**Comments to the Author**

1. If the authors have adequately addressed your comments raised in a previous round of review and you feel that this manuscript is now acceptable for publication, you may indicate that here to bypass the “Comments to the Author” section, enter your conflict of interest statement in the “Confidential to Editor” section, and submit your "Accept" recommendation.

Reviewer #1: All comments have been addressed

2. Is the manuscript technically sound, and do the data support the conclusions?

Reviewer #1: Yes

3. Has the statistical analysis been performed appropriately and rigorously? 

Reviewer #1: Yes

4. Have the authors made all data underlying the findings in their manuscript fully available?

Reviewer #1: Yes

5. Is the manuscript presented in an intelligible fashion and written in standard English?

Reviewer #1: Yes

6. Review Comments to the Author

Reviewer #1: (No Response)

7. PLOS authors have the option to publish the peer review history of their article (what does this mean?). If published, this will include your full peer review and any attached files.

Reviewer #1: No

---

## [Editor Report · Acceptance letter]

5 Apr 2024

PONE-D-23-28150R1 

PLOS ONE

Dear Dr. Millet, 

I'm pleased to inform you that your manuscript has been deemed suitable for publication in PLOS ONE. Congratulations! Your manuscript is now being handed over to our production team.

Kind regards, 

on behalf of

Professor Awatif Abid Al-Judaibi 

Academic Editor

PLOS ONE